# BridgEAD: A Vision-Language Framework for Action Modeling in End-to-End Autonomous Driving

## Abstract

Recently, Vision-Language Models (VLMs) have shown promising prospects in autonomous driving tasks by leveraging rich world knowledge. However, current methods still face significant challenges in aligning the semantic space with the action space and struggle to maintain robust performance in closed-loop evaluations and long-tail scenarios. To address these challenges, we propose BridgEAD in this paper, a novel Vision-Language-Action (VLA) framework for end-to-end autonomous driving that unifies action planning and semantic reasoning. It integrates multi-view visual inputs and historical context into an unmodified VLM backbone for driving scenario reasoning, and leverages a diffusion-based generative planner to further align multimodal scene representations with precise trajectories. We employ supervised fine-tuning for model training to enable end-to-end optimization, thereby endowing BridgEAD with visual question-answering and trajectory planning capabilities. Extensive experiments on multiple benchmarks, including nuScenes, NAVSIM, and Bench2Drive, demonstrate that BridgEAD achieves superior trajectory planning performance in both open-loop and closed-loop evaluations across challenging driving environments. Qualitative results further highlight BridgEAD's strong semantic reasoning ability in driving-related question-answering tasks. We will make our code publicly available upon publication to support future research in this domain.

## 1 Introduction

Autonomous driving is increasingly shifting from modular pipelines which separate perception (Li et al., 2024c; Wang et al., 2022), prediction (Zhou et al., 2023; Huang et al., 2023a), and planning (Huang et al., 2023b; Liu et al., 2025), to end-to-end (E2E) paradigms that map raw sensor inputs directly to driving actions. While modular systems support interpretable engineering and independent module optimization, they are prone to error propagation and cannot be optimized jointly, often resulting in suboptimal overall behavior (Shao et al., 2023; Jia et al., 2023b). E2E approaches address these issues by learning a direct mapping from raw sensor observations to driving actions, typically via imitation of human driving data (Chen et al., 2024b; Hu et al., 2023). However, the predominant E2E methods (Chen et al., 2024a) remain trajectory-driven and lack high-level semantic understanding and causal reasoning, limiting generalization in long-tail, ambiguous, or socially complex scenarios and hindering the diagnosis of failure modes.

Vision–Language Models (VLMs) (Achiam et al., 2023; Alayrac et al., 2022) provide a powerful framework to address the limitations of traditional end-to-end driving approaches by combining high-dimensional visual features with world knowledge and commonsense reasoning encoded in large language models. By integrating perceptual inputs with language-driven priors, VLMs enable decision-making beyond simple trajectory imitation, supporting semantically grounded and causally informed reasoning. The Vision–Language–Action (VLA) (Zhou et al., 2025b;a; Fu et al., 2025) paradigm further extends this capability to action generation, translating abstract scene understanding into executable trajectories. This integration improves both robustness and interpretability, particularly in long-tail driving scenarios (Kim et al., 2024).

Despite these advances, current VLA approaches face two fundamental challenges. First, a persistent semantic-to-action gap (Li et al., 2025c;b)arises because semantic descriptions are inherently ambiguous, while trajectories must satisfy precise kinematic constraints. Discrete or overly abstract intermediate representations often discard critical geometric and dynamic details, preventing reliable alignment between reasoning and control. Second, empirical evaluation remains limited in scope: existing studies typically restrict their evaluation to open-loop (Wang et al., 2024b; Zhou et al., 2025a; Li et al., 2025c; Jiang et al., 2025a) or closed-loop settings, with few providing comprehensive assessments across both paradigms on widely adopted benchmarks such as nuScenes (Caesar et al., 2020), Bench2Drive (Jia et al., 2024), and NAVSIM (Dauner et al., 2024).

Guided by these observations, we propose BridgEAD, an end-to-end VLA framework that integrates unmodified VLM backbones with a diffusion-based generative planner in a plug-and-play design. BridgEAD aligns multimodal scene representations with executable trajectories while remaining lightweight, portable, and easily upgradable as foundation models evolve. To resolve the semantic-to-trajectory mismatch, we introduce a generative action planner that maps the VLM's perception and semantic reasoning output into the trajectory domain, enabling reliable end-to-end planning across diverse and challenging driving environments.

Our main contributions are summarized as follows:

- Plug-and-Play VLA Framework: We introduce BridgEAD, a plug-and-play VLA framework that leverages open-source VLMs for autonomous driving without any architectural modifications, ensuring portability and scalability as foundation models evolve.
- Generative Action Planner: We design a diffusion-based action planner that bridges the semantic–trajectory gap, translating high-level reasoning from VLMs into precise and kinematically feasible trajectories.
- Comprehensive Evaluation: We conduct extensive experiments on nuScenes, NAVSIM, and Bench2Drive, demonstrating that BridgEAD consistently achieves state-of-the-art planning performance in both open-loop and closed-loop settings.

## 2 RELATED WORKS

### 2.1 END-TO-END AUTONOMOUS DRIVING

End-to-end autonomous driving seeks to unify perception, prediction, and planning within a single neural network, alleviating the cascading errors inherent in modular pipelines. Seminal behavioral cloning works (Bojarski et al., 2016) validated the paradigm's feasibility, but their generalization capabilities were constrained by causal confusion and distribution shift. Subsequent efforts to enhance robustness introduced auxiliary guidance through conditional instructions (Codevilla et al., 2018) or leveraged Transformer architectures, such as TransFuser (Chitta et al., 2022), for improved multimodal fusion. More recent approaches enhance performance by integrating tasks into planning-centric frameworks, such as UniAD (Hu et al., 2023), VAD (Jiang et al., 2023), and ParaDrive (Weng et al., 2024), or by employing generative models like GenAD (Zheng et al., 2024) and Diffusion-Drive (Liao et al., 2025) to address trajectory multimodality.

### 2.2 VISION-LANGUAGE MODELS IN AUTONOMOUS DRIVING

To tackle long-tail driving scenarios that remain challenging for conventional end-to-end models, large vision-language models (Wang et al., 2024a) have been explored as a promising paradigm due to their strong commonsense reasoning and scene understanding capabilities. Existing approaches generally fall into two categories. The first category leverages VLMs as high-level decision advisors. For example, DriveVLM (Tian et al., 2024) employs Qwen-VL (Bai et al., 2023) to produce low-frequency waypoints for downstream planning, while Senna (Jiang et al., 2024) integrates Vicuna (Zheng et al., 2023) to generate linguistic "meta-actions" that guide an end-to-end policy. The second category attempts to generate trajectories directly with VLMs, as in EMMA (Hwang et al., 2024), which is built on Gemini (Team et al., 2023), and OmniDrive (Wang et al., 2024b), which adopts a LLaMA-based (Touvron et al., 2023) architecture. However, these methods reveal a fundamental weakness. As most general-purpose VLMs are pre-trained on static web data, they struggle with the dynamic spatiotemporal reasoning required for driving. This limitation creates a critical

Figure 1: The pipeline of our BridgEAD. It consists of three key components: 1) model inputs: a vision encoder projects multi-view inputs into visual tokens and a text tokenizer extracts ego states and navigation commands into language tokens; 2) reasoning space: a language head generates scene question-answering and driving decisions; 3) action space: a diffusion-based planner generates future trajectories.

semantic-to-action gap: the abstract outputs from VLMs cannot be easily translated into precise, kinematically-feasible control signals.

## 2.3 VISION-LANGUAGE-ACTION MODELS

To bridge the semantic-to-action gap, the research community has developed vision-language-action models (Zhou et al., 2025a). Drawing inspiration from robotics (Reed et al., 2022; Brohan et al., 2022; Zitkovich et al., 2023), VLAs directly integrate action generation into the language model's output space. Two primary paradigms have emerged. The first formulates action prediction as a continuous regression problem. However, directly regressing high-dimensional trajectories often yields dynamically infeasible outputs and is susceptible to mode collapse (Chi et al., 2025). The second, increasingly favored paradigm—inspired by seminal works like Gato (Reed et al., 2022) and RT-2 (Zitkovich et al., 2023)—discretizes the continuous action space into discrete tokens that form a learnable action vocabulary. This reframes trajectory planning as a sequence generation task, which is naturally aligned with the auto-regressive nature of language models. Recent works such as ReCogDrive (Li et al., 2025c), AutoVLA (Zhou et al., 2025b), OpenDriveVLA (Zhou et al., 2025a), and Impromptu VLA (Chi et al., 2025) have demonstrated the promise of this paradigm. Nevertheless, existing action vocabularies remain simplistic and often overlook vehicle kinematic priors, which can yield physically infeasible trajectories.

## 3 METHOD

This paper proposes an end-to-end VLA framework for jointly semantic reasoning and action prediction, termed BridgEAD. The pipeline of our model is shown in Fig. 1. Specifically, the multi-view images are first encoded by a pre-trained vision encoder, and the obtained visual tokens are embedded into the language space through linear projections. Then, the current and historical states of the ego-vehicle, as well as the navigation instructions, are encoded into language tokens by the text tokenizer. The large language model (Section 3.1) subsequently combines these features, performing multimodal scene understanding. Finally, a diffusion planner (Section 3.2) is designed to predict the sequence of future actions conditioned on the planning token. The overall model is trained using supervised fine-tuning (Section 3.3), where the parameters of the vision encoder are frozen, while all other parameters are trained.

## 3.1 VISION LANGUAGE MODEL

**Model Inputs.** BridgEAD integrates multi-view visual inputs, high-level navigation commands, and ego-vehicle states to perform scene understanding and trajectory planning jointly. The model

receives RGB streams $I_{cam}$ from six onboard cameras (front, front-left, front-right, back, back-left, back-right), each capturing four consecutive frames at 2 Hz, which are used to encode environment information. High-level navigation instructions $T_{nav}$, such as "*Turn Left*" or "*Go Straight*" explicitly specify the intended route. The ego-vehicle states $T_{ego}$, including the current velocity, acceleration, and the history of the past six frames, provide rich temporal and dynamic information to guide the planning. In addition, we design a planning Question Answering (QA) template with a special planning token $t_{plan}$ for the large language model to accumulate the understanding and reasoning context of the entire driving scenario into the planning token. BridgEAD can perform various text-based understanding and reasoning tasks in the driving scenario, including scene analysis $O_{QA}$, driving decision $O_{dec}$, and action reasoning $O_{traj}$. The overall process is formally written as:

$$O_{QA}, O_{dec}, O_{traj} = \text{BridgEAD}(I_{cam}, T_{nav}, T_{ego}, t_{plan}). \tag{1}$$

**Base VLM.** In this work, we adopt InternVL3-2B (Zhu et al., 2025) as the baseline vision-language model, which offers a good trade-off between efficiency and performance, making it suitable for the autonomous driving system.

## 3.2 Diffusion planner

Generative models are capable of capturing intrinsic data characteristics by learning underlying distributions. Recent studies have shown that latent spaces of different modalities exhibit semantic correlations, such that manipulating the distribution parameters of one modality can influence the generation process. Motivated by these findings, we propose a generative planner to bridge the gap between the VLM's semantic reasoning space and action space. To account for the inherent differences between the reasoning and action distributions, we employ a diffusion model to align them within a Gaussian latent space. Specifically, the proposed diffusion planner takes the planning token $t_{plan}$ along with a series of noisy actions $(a_t^s, a_{t+1}^s, ..., a_{t+N}^s)$ as input, where $s$ denotes the current denoising step and $N$ denotes the length of the output sequence. The model, consisting of several Diffusion Transformer (DiT) (Peebles & Xie, 2023; Li et al., 2024a) blocks, predicts the final action sequences $(a_t, a_{t+1}, ... a_{t+N})$ through multiple denoising steps.

Here, the DiT block incorporates both self-attention and cross-attention mechanisms. Self-attention captures temporal dependencies within the action sequence, ensuring motion continuity. Cross-attention establishes a mapping between the generative condition and the action space. This allows visual, semantic, and historical cues to guide trajectory generation directly.

## 3.3 Training Objectives

We employ Supervised Fine-Tuning (SFT) to train BridgEAD for Visual Question Answering (VQA) and trajectory planning. The model processes multi-view camera inputs, high-level navigation commands, and ego-vehicle states, and produces two outputs: language tokens for VQA and action sequences for trajectory prediction. Training is guided by two complementary objectives: a causal language modeling loss $\mathcal{L}_{llm}$ that strengthens reasoning and comprehension, and an auxiliary action loss $\mathcal{L}_{dit}$ that enhances planning accuracy. Joint optimization of these objectives enables the model to seamlessly integrate semantic understanding with precise action generation in an end-to-end framework. The total loss of the proposed BridgEAD is:

$$\mathcal{L} = \lambda_{llm}\mathcal{L}_{llm} + \lambda_{dit}\mathcal{L}_{dit}, \tag{2}$$

where $\lambda_{llm}$ and $\lambda_{dit}$ are non-negative weighting coefficients that balance the contributions of the LLM loss $\mathcal{L}_{llm}$ and the DiT loss $\mathcal{L}_{dit}$, respectively.

## 4 Experiments

### 4.1 Experimental Setup

#### 4.1.1 Datasets

We train and evaluate BridgEAD's trajectory-planning ability on nuScenes, NAVSIM, and Bench2Drive datasets. For semantic supervision, we use two additional VQA datasets (OmniDrive-nuScenes (Wang et al., 2025) and Chat-B2D (Fu et al., 2025)).

**nuScenes** is a public large-scale dataset for autonomous driving, providing real-world scenarios with open-loop log replay. We follow the official split, utilizing 28,130 training samples. Each scene in the dataset lasts 20s, which contains keyframe annotations at 2 Hz and features multi-sensor coverage, including six cameras. **NAVSIM** is a real-world planning-oriented dataset that achieves a 360-degree panoramic view through eight cameras and integrates LiDAR point clouds generated by five sensors. In our experiments, we subsample 0.15 M frames for training. In **Bench2Drive**, the training set contains 2.0 M fully annotated frames from 13,638 short clips, uniformly covering 44 interactive scenarios, 23 weathers, and 12 towns. The evaluation uses 220 short routes for granular, multi-ability assessment. We subsample 50k frames for training.

**OmniDrive-nuScenes** is a nuScenes-based 3D VQA benchmark covering perception, reasoning, and planning. **Chat-B2D** is a Bench2Drive-aligned VQA set with 2.11 M train and 0.12 M val QA pairs across scene description, critical-object behaviors, meta-driving, and planning.

### 4.1.2 BENCHMARK AND METRICS

We evaluate BridgEAD in both open-loop and closed-loop settings across real-world and simulated environments. The open-loop nuScenes employs L2 Distance and Collision Rate as evaluation metrics. The non-reactive NAVSIM leverages Predictive Driver Model Score (PDMS) to measure key driving behaviors, including No at-fault Collision (NC), Drivable Area Compliance (DAC), Ego Progress (EP), Time-to-Collision (TTC), and Comfort (C). For the closed-loop Bench2Drive benchmark in the CARLA simulator, driving performance is assessed using Driving Score (DS), Success Rate (SR), Efficiency, and Comfortness. Additionally, we employ BLEU (Papineni et al., 2002), ROUGE (Lin, 2004), and CIDEr (Vedantam et al., 2015) to evaluate the performance of BridgEAD on semantic reasoning tasks.

### 4.1.3 IMPLEMENTATION DETAILS

All experiments are conducted on 32 NVIDIA A800 GPUs with 80 GB of memory. We use a per-GPU batch size of 1 and accumulate gradients, resulting in an effective batch size of 128. We use AdamW with a learning rate of $4 \times 10^{-5}$. In the loss function, the weighting parameter $\lambda_{dit}$ is set to 1.0, $\lambda_{llm}$ decreases exponentially from 1.0 to 0.05 with training steps.

### 4.2 MAIN RESULTS

Table 1 presents the results of BridgEAD compared with existing end-to-end and VLM-based methods on the NAVSIM benchmark. BridgEAD achieves a 2.98% higher PDMS than Transfuser, despite utilizing both camera and LiDAR inputs. Among VLM-based SFT approaches, BridgEAD surpasses AutoVLA by 7.45% in PDMS, and achieves a PDMS comparable to that of the 8B ReCogDrive model with only 2B parameters.

As shown in Table 2 and Table 3, BridgEAD significantly exceeds all end-to-end methods in closed-loop metrics on the Bench2Drive benchmark and achieves performance comparable with other VLM-based methods. Specifically, it achieves the highest merging ability with a score of 31.25, and ranks among the top models with a Driving Score of 76.52 and a Success Rate of 51.36.

Additionally, Table 4 illustrates the results on the nuScenes benchmark. BridgEAD shows significant advantages in trajectory prediction accuracy and safety, achieving an average L2 distance of 0.37 meters and the lowest average collision rate of 0.1% among the VLM-based methods. These results highlight BridgEAD's robustness and long-term decision-making safety in complex dynamic environments.

As shown in Table 5, BridgEAD demonstrates competitive performance on the driving scene VQA tasks. When temporal questions are excluded, BridgEAD attains a high BLEU score of 52.31 in the Chat-B2D benchmark. In the OminiDrive-nuScenes benchmark, BridgEAD obtains the best VQA reasoning ability, with a CIDEr score of 85.68, surpassing OmniDrive-Q by 17.08 and OmniDrive-L by 12.48.

Table 1: Performance comparison on the NAVSIM Navtest benchmark. C/L refers to camera/LiDAR. RFT: Reinforcement Fine-Tuning.

| Method | Modality | NC ↑ | DAC ↑ | EP ↑ | TTC ↑ | C ↑ | PDMS ↑ |
|---|---|---|---|---|---|---|---|
| End-to-end Methods | | | | | | | |
| Ego Status MLP | – | 93.0 | 77.3 | 62.8 | 83.6 | **100.0** | 65.6 |
| LTF (Chitta et al., 2022) | C | 97.4 | 92.8 | 79.0 | 92.4 | **100.0** | 83.8 |
| Transfuser (Chitta et al., 2022) | C&L | 97.7 | 92.8 | 79.2 | 92.8 | **100.0** | 84.0 |
| DiffusionDrive (Liao et al., 2025) | C&L | 98.2 | 96.2 | 82.2 | 94.7 | **100.0** | 88.1 |
| Hydra-MDP (Li et al., 2024b) | C&L | 99.1 | **98.3** | 85.2 | 96.6 | **100.0** | 91.3 |
| TrajHF (Li et al., 2025a) | C&L | 99.3 | 97.5 | 90.4 | **98.0** | 99.8 | 94.0 |
| TransDiffuser (Jiang et al., 2025b) | C&L | **99.4** | 96.5 | **94.1** | 97.8 | 99.4 | **94.9** |
| VLM-based Methods | | | | | | | |
| AutoVLA (SFT) (Zhou et al., 2025b) | C | 96.9 | 92.4 | 75.8 | 88.1 | 99.9 | 80.5 |
| AutoVLA (RFT) (Zhou et al., 2025b) | C | 98.4 | 95.6 | 81.9 | 98.0 | 99.9 | 89.1 |
| ReCogDrive (SFT) (Li et al., 2025c) | C | 98.3 | 95.1 | 81.1 | 94.3 | **100.0** | 86.8 |
| ReCogDrive (RFT) (Li et al., 2025c) | C | 98.2 | 97.8 | 85.3 | 95.2 | 99.8 | 89.6 |
| BridgEAD (Ours) | C | 98.0 | 96.2 | 81.0 | 93.3 | 98.1 | 86.5 |

Table 2: Closed-loop and Open-loop performance comparison on the Bench2Drive benchmark. C/L refers to camera/LiDAR. L2 is averaged over the predictions in 2s under 2Hz.

| Method | Modality | Open-loop Metric L2 (m) ↓ | Close-loop Metric | | | |
|---|---|---|---|---|---|---|
| | | | DS ↑ | SR (%) ↑ | Efficiency ↑ | Comfortness ↑ |
| End-to-end Methods | | | | | | |
| AD-MLP (Zhai et al., 2023) | C | 3.64 | 18.05 | 0.00 | 48.45 | 22.63 |
| UniAD-Tiny (Hu et al., 2023) | C | 0.80 | 40.73 | 13.18 | 123.92 | **47.04** |
| VAD (Jiang et al., 2023) | C | 0.91 | 42.35 | 15.00 | **157.94** | 46.01 |
| UniAD-Base (Hu et al., 2023) | C | 0.73 | 45.81 | 16.36 | 129.21 | 43.58 |
| TCP-traj (Wu et al., 2022) | C | 1.70 | 59.90 | 30.00 | 76.54 | 18.08 |
| ThinkTwice (Jia et al., 2023b) | C | 0.95 | 62.44 | 31.23 | 69.33 | 16.22 |
| DriveTransformer-Large (Jia et al., 2025) | C | **0.62** | 63.46 | 35.01 | 100.64 | 20.78 |
| DriveAdapter (Jia et al., 2023a) | C&L | 1.01 | 64.22 | 33.08 | 70.22 | 16.01 |
| VLM-based Methods | | | | | | |
| Orion (Fu et al., 2025) | C | 0.68 | 77.74 | 54.62 | 151.48 | 17.38 |
| AutoVLA (Zhou et al., 2025b) | C | – | **78.84** | **57.73** | 146.93 | 39.33 |
| BridgEAD (Ours) | C | 0.98 | 76.52 | 51.36 | 146.11 | 7.82 |

Table 3: Multi-Ability performance comparison on the Bench2Drive benchmark. C/L refers to camera/LiDAR.

| Method | Modality | Ability (%) ↑ | | | | | |
|---|---|---|---|---|---|---|---|
| | | Merging | Overtaking | Emergency Brake | Give Way | Traffic Sign | Mean |
| End-to-end Methods | | | | | | | |
| AD-MLP (Zhai et al., 2023) | C | 0.00 | 0.00 | 0.00 | 0.00 | 4.35 | 0.87 |
| TCP-ctrl* (Wu et al., 2022) | C | 10.29 | 4.44 | 10.00 | 10.00 | 6.45 | 8.23 |
| TCP* (Wu et al., 2022) | C | 16.18 | 20.00 | 20.00 | 10.00 | 6.99 | 14.63 |
| UniAD-Tiny (Hu et al., 2023) | C | 8.89 | 9.33 | 20.00 | 20.00 | 15.43 | 14.73 |
| UniAD-Base (Hu et al., 2023) | C | 14.10 | 17.78 | 21.67 | 10.00 | 14.21 | 15.55 |
| VAD (Jiang et al., 2023) | C | 8.11 | 24.44 | 18.64 | 20.00 | 19.15 | 18.07 |
| TCP-traj* (Wu et al., 2022) | C | 8.29 | 24.29 | 51.67 | 40.00 | 46.28 | 34.22 |
| ThinkTwice (Jia et al., 2023b) | C | 27.38 | 18.42 | 35.52 | **50.00** | 54.23 | 37.17 |
| DriveTransformer-Large (Jia et al., 2025) | C | 17.57 | 35.00 | 48.36 | 40.00 | 52.10 | 38.60 |
| DriveAdapter (Jia et al., 2023a) | C&L | 28.82 | 26.38 | 48.76 | **50.00** | 56.43 | 42.08 |
| VLM-based Methods | | | | | | | |
| Orion (Fu et al., 2025) | C | 25.00 | **71.11** | **78.33** | 30.00 | **69.15** | **54.72** |
| BridgEAD (Ours) | C | **31.25** | 57.78 | 75.00 | 40.00 | 67.37 | 54.28 |

Table 4: Open-loop performance comparison on the nuScenes benchmark. L2 and Collision Rate are evaluated under ST-P3 metrics.

| Method | Ego Status | | L2 (m) ↓ | | | | Collision (%) ↓ | | | |
|---|---|---|---|---|---|---|---|---|---|---|
| | BEV | Planner | 1s | 2s | 3s | Avg. | 1s | 2s | 3s | Avg. |
| **End-to-end Methods** | | | | | | | | | | |
| Ego-MLP | – | ✓ | 0.15 | 0.32 | 0.59 | 0.35 | **0.00** | 0.27 | 0.85 | 0.37 |
| ST-P3 (Hu et al., 2022) | – | – | 1.33 | 2.11 | 2.90 | 2.11 | 0.23 | 0.62 | 1.27 | 0.71 |
| UniAD (Hu et al., 2023) | – | – | 0.48 | 0.96 | 1.65 | 1.03 | 0.05 | 0.17 | 0.71 | 0.31 |
| UniAD (Hu et al., 2023) | ✓ | ✓ | 0.20 | 0.42 | 0.75 | 0.46 | 0.02 | 0.25 | 0.84 | 0.37 |
| VAD (Jiang et al., 2023) | ✓ | – | 0.41 | 0.70 | 1.06 | 0.72 | 0.04 | 0.43 | 1.15 | 0.54 |
| VAD (Jiang et al., 2023) | ✓ | ✓ | 0.17 | 0.34 | 0.60 | 0.37 | 0.04 | 0.27 | 0.67 | 0.33 |
| SparseDrive (Sun et al., 2025) | – | ✓ | 0.29 | 0.58 | 0.96 | 0.61 | 0.01 | **0.05** | 0.18 | **0.08** |
| DiffusionDrive (Liao et al., 2025) | ✓ | ✓ | 0.27 | 0.54 | 0.90 | 0.57 | 0.03 | **0.05** | **0.16** | **0.08** |
| **VLM-based Methods** | | | | | | | | | | |
| DriveVLM (Tian et al., 2024) | – | – | 0.18 | 0.34 | 0.68 | 0.40 | 0.10 | 0.22 | 0.45 | 0.27 |
| OmniDrive (Wang et al., 2025) | – | – | 0.40 | 0.80 | 1.32 | 0.84 | 0.04 | 0.46 | 2.32 | 0.94 |
| OmniDrive++ (Wang et al., 2025) | ✓ | ✓ | **0.14** | **0.29** | 0.55 | 0.33 | **0.00** | 0.13 | 0.78 | 0.30 |
| EMMA (Hwang et al., 2024) | – | – | **0.14** | **0.29** | **0.54** | **0.32** | – | – | – | – |
| ORION (Fu et al., 2025) | ✓ | – | 0.17 | 0.31 | 0.55 | 0.34 | 0.05 | 0.25 | 0.80 | 0.37 |
| OpenDriveVLA (Zhou et al., 2025a) | ✓ | ✓ | **0.14** | 0.30 | 0.55 | 0.33 | 0.02 | 0.07 | 0.22 | 0.10 |
| AutoVLA (Zhou et al., 2025b) | – | ✓ | 0.25 | 0.46 | 0.73 | 0.48 | 0.07 | 0.07 | 0.26 | 0.13 |
| BridgEAD (Ours) | – | ✓ | 0.17 | 0.32 | 0.61 | 0.37 | 0.04 | 0.10 | 0.17 | 0.10 |

Table 5: Performance comparison of semantic reasoning ability on the OmniDrive-nuScenes and Chat-B2D benchmarks. (%)

| Method | Chat-B2D | | | OmniDrive-nuScenes | |
|---|---|---|---|---|---|
| | BLEU ↑ | ROUGE ↑ | CIDEr ↑ | ROUGE ↑ | CIDEr ↑ |
| OmniDrive-Q (Wang et al., 2025) | - | - | - | 32.60 | 68.60 |
| OmniDrive-L (Wang et al., 2025) | - | - | - | - | 73.20 |
| ORION (VQA FT) (Fu et al., 2025) | 50.82 | **77.65** | 65.65 | - | - |
| ORION (SFT) (Fu et al., 2025) | **52.49** | 77.58 | **65.77** | - | - |
| BridgEAD (Ours) | 52.31 | 57.15 | 38.25 | **36.51** | **85.68** |

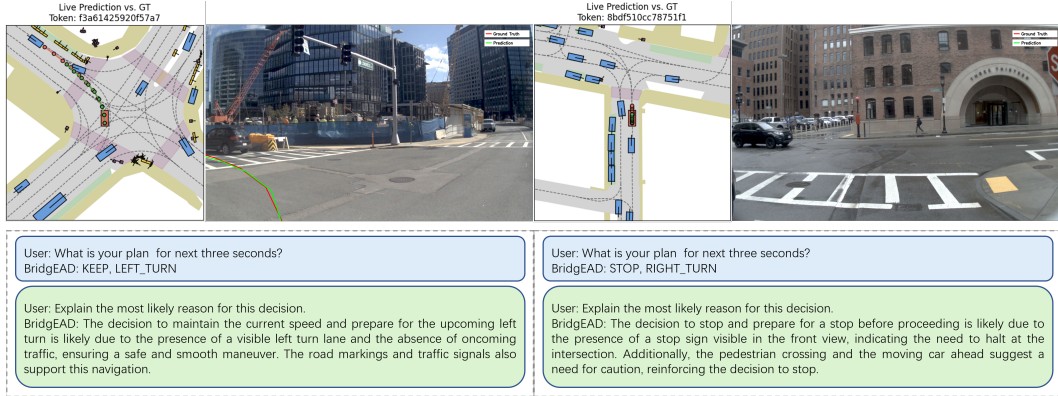

Figure 2: Planning and QA results of BridgEAD on the NAVSIM benchmark. Our model can make correct speed and behavior decisions while providing reasonable explanations in scenarios involving a left turn at a signalized intersection and a right turn at a stop sign.

Figure 3: Planning and QA Results of BridgEAD on the nuScenes benchmark. By analyzing the surrounding images, our model identifies and localizes multiple crossing pedestrians, and consequently infers the driving decision to maintain a low speed while proceeding straight.

### 4.3 QUALITATIVE RESULTS

Fig. 2 and Fig. 3 present qualitative results of BridgEAD on the NAVSIM and nuScenes benchmarks, showcasing both its planning trajectories and reasoning capabilities. The model's question-and-answer outputs reveal a sophisticated understanding of the driving environment, demonstrating its ability to perceive complex scenarios and identify salient objects accurately. This perceptual awareness allows the model to form logical inferences that inform its planning decisions. Consequently, the generated trajectories are highly consistent with the ground truth. For example, upon detecting a stop sign at a right-turn intersection, the model decides to stop and yield; at the entrance to an industrial area, by accurately recognizing the positions of traffic cones and pedestrians, the model chooses to proceed slowly to ensure driving safety. These results collectively demonstrate the reliability and safety of our model in dynamic and uncertain environments.

### 4.4 ABLATION STUDY

We conduct an ablation study of the trajectory generation method. Several previous works (Seff et al., 2023; Sima et al., 2024; Zhou et al., 2025b) benefit from the action token approach, which involves the model outputting a sequence of action tokens that are subsequently decoded into a planning trajectory using a predefined action codebook. Other works (Xie et al., 2025; Wang et al., 2025; Hwang et al., 2024) employ the text number method, requiring a subsequent parsing step to extract the numerical values from the model's textual output. As demonstrated in Table 6, our DiT method significantly outperforms the action token and text number methods. Furthermore, the action token and text number methods incur higher time costs. The text number method also suffers from instability in model generation due to the dependency on coordinate extraction from the generated text.

Table 6: Ablation study on different trajectory generation methods. The closed-loop metric on the Bench2Drive benchmark is evaluated on Route Dev10.

| Planning Method | NAVSIM | Bench2Drive | | | nuScenes | |
|---|---|---|---|---|---|---|
| | PDMS ↑ | L2 (m) ↓ | DS ↑ | SR (%) ↑ | L2 (m) ↓ | Collision (%) ↓ |
| DiT | **86.50** | **0.98** | **69.63** | **30.00** | **0.37** | **0.10** |
| Text number | 81.70 | 1.07 | 39.90 | 20.00 | **0.37** | 0.13 |
| Action token | 70.35 | 1.08 | 22.93 | 0.00 | 0.40 | 0.29 |

## 5 CONCLUSION

In this work, we propose BridgEAD, a novel vision-language-action model for unified action planning and semantic reasoning in autonomous driving scenarios. We employ an unmodified VLM backbone to obtain multimodal representations from multi-view visual inputs and historical context. We then design a diffusion-based action planner to better align high-level semantic reasoning with precise trajectory generation. Finally, we apply supervised fine-tuning to enhance the model's integrated capabilities in visual question-answering and trajectory planning. Extensive experiments on nuScenes, NAVSIM, and Bench2Drive demonstrate that BridgEAD achieves competitive performance on both open-loop and closed-loop planning benchmarks and exhibits robust semantic reasoning capabilities.

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
