# OpenReview forum: "BridgEAD: A Vision-Language Framework for Action Modeling in End-to-End Autonomous Driving"
_ICLR.cc/2026/Conference — Submitted to ICLR 2026_

### Official Review · Reviewer_41o2 · 2025-10-30

**Soundness:** 3
**Presentation:** 2
**Contribution:** 2
**Rating:** 2
**Confidence:** 5

**Summary:**

This paper proposes to integrate a diffusion based generative planner module with a VLM backbone processing scene images and navigational text. They train the model end to end with supervised fine-tuning while aligning multi-modal scene representations with trajectories. They run evaluations on NuScenes, NAVSIM and Bench2Drive showing comparable performance with SOTA models.

**Strengths:**

1. VLM does not need any fine-tuning. Any public VLM can be used for this architecture in a plug and play manner.
2. Planning token is a learnable and continuous, without any action code-book or dictionary. This is more flexible as the planning token remains latent, without requiring any supervision or explicit interpretation.

**Weaknesses:**

1. Performance. In the tables presented closed and open loop performance is worse than other VLM based approaches. There are specific abilities where this approach performs better, but overall there is performance degradation.
2. Using diffusion for planner is expensive, there should be some discussion on latency of such method.
3. Use of generic VLM as frozen can also be seen as weakness, as these models can have limited understanding of driving maneuvers. It is worth experimenting with VLMs fine-tuned on driving data like OmniDrive.

**Questions:**

1. The outputs of language head and planner head are used for separate loss functions, there is no direct alignment between these outputs. So, it's possible that the language model producing driving decision inconsistent with the planner. Did the authors consider any mechanism to prevent this?
2. The authors claim the trajectories generated are kinematically feasible. How was this enforced? The conditioning of the diffusion model with planner token does not enforce vehicle kinematic constraints.

---

### Official Review · Reviewer_UwNe · 2025-11-01

**Soundness:** 2
**Presentation:** 3
**Contribution:** 2
**Rating:** 4
**Confidence:** 4

**Summary:**

BridgEAD targets a pivotal gap in VLA-based autonomous driving—the "semantic-to-action gap"—by leveraging an unmodified VLM for semantic reasoning and a diffusion planner for trajectory generation, demonstrating practical value in its modular design and comprehensive evaluation across open/closed-loop settings. The paper’s coverage of diverse datasets (real-world nuScenes, simulation-based Bench2Drive) and ablation studies on trajectory generation methods provide partial support for its core claims. However, critical limitations undermine its impact: key technical details (e.g., kinematic constraint integration in the diffusion planner) are underspecified, evaluations lack dedicated validation for long-tail scenarios, and most importantly, the model fails to achieve state-of-the-art (SOTA) performance across key benchmarks, with results either matching or falling short of existing baselines. Clarifying technical mechanisms, expanding evaluation scope, and acknowledging performance boundaries would strengthen the work’s contribution to VLA-driven autonomous driving.

**Strengths:**

The paper clearly identifies the dual limitations of current end-to-end (E2E) and VLM-based autonomous driving: traditional E2E methods lack high-level semantic reasoning, limiting generalization in long-tail scenarios; existing VLM-based approaches struggle with aligning abstract semantic outputs to precise, kinematically feasible trajectories. BridgEAD’s split design—retaining the VLM’s semantic reasoning via unmodified integration while using a diffusion planner for trajectory specialization—directly addresses this semantic-to-action gap.
Plug-and-Play Modularity： By freezing the pre-trained VLM backbone and adding a separate diffusion planner, BridgEAD avoids the complexity of modifying VLM architectures. This design enables easy upgrades as foundation VLMs evolve and reduces deployment costs, while lightweight interfaces (linear projection for visual tokens, text tokenizer for ego states/instructions) facilitate efficient multimodal fusion .
Comprehensive Evaluation：Unlike many works that focus solely on open-loop evaluations, BridgEAD is validated on three distinct benchmarks (nuScenes, NAVSIM, Bench2Drive) covering real-world and simulated environments. It assesses both trajectory planning (e.g., L2 distance, collision rate in open-loop; driving score, success rate in closed-loop) and semantic reasoning (VQA via BLEU, ROUGE, CIDEr), providing a holistic view of its VLA capabilities

**Weaknesses:**

While the paper claims the diffusion planner generates "kinematically feasible trajectories," it provides no details on how vehicle dynamics (e.g., maximum acceleration, steering angle limits) are integrated.

Despite framing long-tail scenario performance as a key motivation, the paper does not conduct dedicated evaluations for extreme conditions (e.g., heavy rain, severe occlusion) or dynamic multi-agent interactions (e.g., unprotected left turns, sudden pedestrian crossings). Bench2Drive’s 23 weather conditions and 44 scenarios are only reported in aggregate, with no analysis of performance degradation in low-probability, high-risk scenarios

The paper fails to clarify how the VLM’s semantic outputs, driving decisions guide the diffusion planner.

BridgEAD does not achieve SOTA results on any major benchmark.

**Questions:**

While the paper claims the diffusion planner generates "kinematically feasible trajectories," it provides no details on how vehicle dynamics (e.g., maximum acceleration, steering angle limits) are integrated.

Despite framing long-tail scenario performance as a key motivation, the paper does not conduct dedicated evaluations for extreme conditions (e.g., heavy rain, severe occlusion) or dynamic multi-agent interactions (e.g., unprotected left turns, sudden pedestrian crossings). Bench2Drive’s 23 weather conditions and 44 scenarios are only reported in aggregate, with no analysis of performance degradation in low-probability, high-risk scenarios

The paper fails to clarify how the VLM’s semantic outputs, driving decisions guide the diffusion planner.

BridgEAD does not achieve SOTA results on any major benchmark.

---

### Official Review · Reviewer_mod9 · 2025-11-01

**Soundness:** 2
**Presentation:** 2
**Contribution:** 2
**Rating:** 4
**Confidence:** 4

**Summary:**

This paper proposes BridgEAD, a Vision–Language–Action (VLA) framework to address end-to-end autonomous driving which combines semantic reasoning and trajectory planning. Unlike traditional end-to-end models that lack high-level reasoning, BridgEAD integrates:

- Multi-view visual inputs, navigation commands, and ego-vehicle states into an unmodified Vision-Language Model (VLM) backbone for scene understanding and reasoning.
- A diffusion-based generative planner to bridge the semantic-to-action gap, translating reasoning outputs into precise, kinematically feasible trajectories.
- Supervised fine-tuning for joint optimization of visual question answering (VQA) and trajectory planning.

**Strengths:**

- BridgEAD addresses the semantic-to-action gap effectively, and improving alignment between reasoning and control.
- BridgEAD can be applied in plug-and play option, where BridgEAD uses unmodified VLM backbones, making the approach portable and scalable as foundation models evolve.
- BridgEAD conducted a good ablation study, showing clear comparison of trajectory generation methods (DiT vs. action tokens vs. text numbers), highlighting the superiority of the proposed approach.

**Weaknesses:**

- Despite improvements using the diffusion-based approach still incurs higher inference latency compared to VLP, DiMA which distills VLM knowledge to simpler planners. Comparisons with these relevant methods like VLP and DiMA are missing.
- Comparison scope: Strong against VLM-based baselines, but limited discussion on reinforcement learning or hybrid approaches like Diffusion Planner.
- BridgEAD claims Generative Action Planner as novelty, but integration of diffusion planner or VAE based planner with VLM is explored in ORION.
- BridgEAD Relies on short history (6 frames at 2 Hz); may struggle in highly dynamic or multi-agent scenarios. It would be better if authors can provide an comparison showing the BridgEAD' s temporal reasoning limitations.

- VLP, VLP: Vision Language Planning for Autonomous Driving, CVPR 2024.

- DiMA, Distilling Multi-modal Large Language Models for Autonomous Driving, CVPR 2025.

**Questions:**

- How does BridgEAD handle ambiguity or occlusions in multi-view inputs, especially for critical objects like pedestrians?
- Could temporal context extension (e.g., longer history or predictive features) improve performance in complex scenarios?
- What is the inference latency for closed-loop planning? Is real-time deployment feasible?
- How robust is the diffusion planner to distribution shifts (e.g., different cities, weather, or sensor setups)?
- What are the most common failure cases observed during evaluation, and how severe are they in terms of safety?

---

### Official Review · Reviewer_SJbG · 2025-11-03

**Soundness:** 3
**Presentation:** 3
**Contribution:** 2
**Rating:** 4
**Confidence:** 5

**Summary:**

This paper introduces BridgEAD, which is a vision-language based model for end-to-end driving.  BridgEAD uses language head to perform autoregressive prediction for reasoning and diffusion head for driving action prediction. It achieves good performance on both nuScenes and NAVSIM.

**Strengths:**

1. The idea that uses language autoregressive sampling for reasoning and diffusion model for action prediction is well-motivated and promising.
2. The paper is well-written with good clarity.

**Weaknesses:**

I have major concerns in the experiment results.
1. The authors ignored a new important E2E driving benchmark -- WOD-E2E[1]. This benchmark is designed for long-tailed E2E driving scenarios, which can test model's generalization capability.
2. BridgeEAD did not achieve the best performance at any benchmark. In NAVSIM, it is worse than TransDiffuser. In Bench2Drive, it is worse than AutoVLA. In nuScenes, it is worse than EMMA.
3. A lot of details are not covered by the paper, including but not limited to:
1) Which vlm model did the authors use? Is it Qwen?
2) What are the training configurations for the model?
3) How the latency looks like?

[1] WOD-E2E: Waymo Open Dataset for End-to-End Driving in Challenging Long-tail Scenarios

**Questions:**

To change my opinion, the authors need to :
1) Show results in WOD-E2E.
2) Have a good explanation  of the inferior results in the several benchmarks.
3) Clarify the details.

---

### Meta-Review · Area_Chair_VTs8 · 2026-01-13

**Summary:**

The reviewers agree that the paper presents a clearly written VLA-style end-to-end driving framework that combines an unmodified VLM backbone for semantic reasoning with a diffusion-based planner for trajectory generation, and they find the overall direction promising and reasonably motivated. However, the reviewer concerns that most strongly inform the decision are about empirical competitiveness and validation scope: multiple reviewers point out that BridgEAD does not convincingly lead on key benchmarks relative to contemporaneous baselines (e.g., TransDiffuser / AutoVLA / EMMA) and that the long-tail motivation is not substantiated with targeted evaluation (e.g., WOD-E2E or fine-grained long-tail breakdowns on Bench2Drive), making it difficult to support the paper’s central claims about robustness in challenging scenarios. In addition, the paper is viewed as under-specified in several critical technical aspects (e.g., how kinematic feasibility is enforced in the diffusion planner, how language decisions condition or align with the planner outputs, what VLM backbone/training configuration is used, and what the inference latency looks like), and several reviewers also note missing comparisons to closely related planning/distillation approaches (e.g., VLP, DiMA) and related diffusion/VAE-style integrations (e.g., ORION), which further weakens the contribution case. With one strong reject and three marginal-below-threshold scores, and without a rebuttal/discussion record that would justify expected score flips, the overall signal is not sufficient to clear a selective acceptance bar.

**Reviewer Concerns:**

Regarding rebuttal impact, the provided review package does not include an author rebuttal or any substantive post-review discussion to evaluate whether the authors addressed the above concerns, so it would be inappropriate to infer increased confidence or score movement. As a result, the concerns remain outstanding: missing long-tail evaluation (including the explicit request for WOD-E2E), lack of a convincing explanation for not leading on major benchmarks, missing/unclear technical details around kinematic feasibility and semantic-to-action conditioning, and the limited comparison scope to several relevant VLM-planning/distillation baselines and related integrations. While the paper’s strengths (clarity, modularity, and broad benchmark coverage across nuScenes/NAVSIM/Bench2Drive) are acknowledged, they were not sufficient in the initial reviews to overcome the impact/novelty/validation concerns, and there is no evidence in the record that these were resolved.

**Reviewer Scores:**

Reviewer SJbG (initial 4) is confident and requests targeted additions (notably WOD-E2E), stronger justification for inferior benchmark results, and missing implementation/latency details; absent rebuttal/discussion evidence, the most likely outcome is that this reviewer remains at 4.

Reviewer mod9 (initial 4) emphasizes missing comparisons to VLP/DiMA and novelty positioning vs ORION as well as latency and robustness questions; without follow-up, this is expected to remain 4.

Reviewer UwNe (initial 4) raises underspecification of kinematic constraints, lack of dedicated long-tail analysis, unclear semantic-to-planner interface, and the non-SOTA position; without clarifications, this is expected to remain 4.

Reviewer 41o2 (initial 2) focuses on performance degradation versus other VLM-based approaches, diffusion cost/latency, and potential misalignment between the language head and planner head; given the certainty of this review and no rebuttal evidence, this is expected to remain 2.

---

### Decision · Program_Chairs · 2026-01-26

Reject